# Public-Key Cryptography Based on Tropical Circular Matrices

Huawei Huang [1,*], Chunhua Li [2] and Lunzhi Deng [1]

1   School of Mathematical Sciences, Guizhou Normal University, Guiyang 550025, China; denglunzhi@163.com
2   School of Science, East China Jiaotong University, Nanchang 330013, China; chunhuali66@163.com
*   Correspondence: 201307045@gznu.edu.cn; Tel.: +86-177-8415-1752

**Abstract:** Some public-key cryptosystems based on the tropical semiring have been proposed in recent years because of their increased efficiency, since the multiplication is actually an ordinary addition of numbers and there is no ordinary multiplication of numbers in the tropical semiring. However, most of these tropical cryptosystems have security defects because they adopt a public matrix to construct commutative semirings. This paper proposes new public-key cryptosystems based on tropical circular matrices. The security of the cryptosystems relies on the NP-hard problem of solving tropical nonlinear systems of integers. Since the used commutative semiring of circular matrices cannot be expressed by a known matrix, the cryptosystems can resist KU attacks. There is no tropical matrix addition operation in the cryptosystem, and it can resist RM attacks. The new cryptosystems can be considered as a potential post-quantum cryptosystem.

**Keywords:** cryptographic algorithm; key exchange protocol; public-key encryption scheme; tropical algebra; tropical circular matrices

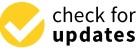



## 1. Introduction

Public-key cryptography was introduced by Diffie and Hellman [1]. In a public-key cryptosystem, the key for encryption is public and the key for decryption is private. Since then, public-key cryptography has been booming and has been widely used in modern communications. Modern public-key cryptography relies mainly on the integer factorization problem (IFP) [2] and discrete logarithm problem (DLP) [1,3]. However, Shor [4] proposed a quantum algorithm that can solve the integer factorization problem and discrete logarithm problem in polynomial time on a quantum computer. So, it is a research area focused on public-key cryptography to design public-key cryptosystems that can resist quantum attacks [5].

In the past two decades, different algebraic structures have been recommended to improve the existing public-key cryptosystems. Some researchers considered non-abelian groups to design public-key cryptosystems such as matrix groups [6–9], braid groups [10,11], inner automorphism groups [12], and ring structures [13] for cryptographic primitives. However, many successful attacks on such cryptosystems have been published [14–17].

Maze, Monico, and Rosenthal proposed one of the first cryptosystems based on semigroups and semirings [18], using some ideas from [10], as well as from their previous article [19]. However, it was broken by Steinwandt et al. [20]. Atani published a cryptosystem using semimodules over factor semirings [21]. Durcheva applied some idempotent semirings to construct cryptographic protocols [22]. A survey on semirings and their cryptographic applications was carried out by Durcheva [23].

Grigoriev and Shpilrain proved that the problem of solving the systems of min-plus polynomial equations in tropical algebra is NP-hard and suggested using a min-plus (tropical) semiring to design a public-key cryptosystem [24]. An obvious advantage of using tropical algebras as platforms is high efficiency because, in tropical schemes, one does not have to perform any multiplication of numbers since tropical multiplication is the

usual addition. However, "tropical powers" of an element exhibit some patterns, even if such an element is a matrix over a tropical algebra. This weakness was exploited by Kotov and Ushakov to propose a fairly successful attack on the public-key cryptosystem in [25]. Then, Grigoriev and Shpilrain improved the original scheme and proposed the public-key cryptosystems based on the semi-direct product of the tropical matrix semiring [26]. However, some attacks on the improved public-key cryptosystem have been suggested by Rudy and Monico [27] and Isaac and Kahrobei [28]. As we know, most of these tropical public-key cryptosystems have security defects because they adopt a public matrix to construct commutative semirings or there is a tropical matrix addition operation in the cryptosystems. A review of the tropical approach in cryptography was carried out by Ahmed, Pal and Mohan [29].

**Our contribution:** This paper provides new public-key cryptosystems based on tropical *t*-circular matrices. The security of the cryptosystem relies on the NP-hard problem of solving tropical nonlinear systems of integers. Since the used commutative semirings of circular matrices cannot be represented by a known matrix and there is no tropical matrix addition operation in the cryptosystem, these cryptosystems can resist all known attacks such as KU attacks and RM attacks. Our results show that these cryptosystems are secure when the computational two-side tropical circular matrices action problem (CTCMAP) and the decisional two-side tropical circular matrices action problem (DTCMAP) are hard. It seems that our cryptosystems based on tropical circular matrices can be considered as potential post-quantum cryptosystems.

The rest of the paper is organized as follows: We focus on some definitions as fundamental key notions of tropical matrix algebra in Section 2. In Section 3, we present the new public-key cryptosystems based on tropical circular matrices. Then, in Section 4, parameter selection and efficiency of the cryptosystems are discussed. Finally, the conclusion and further research are given in Section 5.

## 2. Tropical Matrix Semiring over Integer

The definition of a semiring was first given by Vandiver [30]. These are structures that satisfy all the properties of a ring, except for the existence of additive inverses. Imre Simon, a Brazilian mathematician and computer scientist, discovered what is now known as the tropical semiring [31].

**Definition 1.** ([32]) *Let R be a non-empty set with binary operations "+" and "·"; then, R is called a semiring if it satisfies the following conditions:*

(1)  $(R, +)$ *is a commutative semigroup with an identity element 0;*
(2)  $(R, \cdot)$ *is a semigroup with an identity element $1 \neq 0$;*
(3)  *Multiplication satisfies the left and right distribution law for addition;*
(4)  $(\forall a \in R)\, a \cdot 0 = 0 \cdot a = 0.$

*If $(R, \cdot)$ is commutative, then the semiring is called a commutative semiring.*

**Definition 2.** ([24]) *The integer tropical commutative semiring is the set $\mathcal{Z} = \mathbb{Z} \cup \{\infty\}$ with addition and multiplication as follows:*

$$(\forall x, y \in \mathbb{Z})\, x \oplus y = \min(x, y), \quad x \otimes y = x + y.$$

*$\infty$ satisfies the following equations:*

$$(\forall x \in \mathbb{Z})\, \infty \oplus x = x, \quad \infty \otimes x = \infty.$$

*It is clear that $(\mathcal{Z}, \oplus, \otimes)$ is a commutative semiring whose zero element and unitary element are $\infty$ and 0, respectively.*

Let $M_k(\mathcal{Z})$ be the set of all $k \times k$ matrices over $\mathcal{Z}$. We can also define the tropical matrix $\oplus$ and $\otimes$ operations.

$$\left(\forall A = (a_{ij})_{k \times k}, B = (b_{ij})_{k \times k} \in M_k(\mathcal{Z})\right) A \oplus B = (a_{ij} \oplus b_{ij})_{k \times k}, \ A \otimes B = \left(\sum_{l=1}^{n} a_{il} \otimes b_{lj}\right)_{k \times k}$$

**Example 1.**

$$\begin{pmatrix} 4 & -5 \\ 27 & 0 \end{pmatrix} \oplus \begin{pmatrix} 10 & 3 \\ 1 & 9 \end{pmatrix} = \begin{pmatrix} 4 & -5 \\ 1 & 0 \end{pmatrix}$$

$$\begin{pmatrix} 4 & -5 \\ 27 & 0 \end{pmatrix} \otimes \begin{pmatrix} 10 & 3 \\ 1 & 9 \end{pmatrix} = \begin{pmatrix} -4 & 4 \\ 1 & 9 \end{pmatrix}$$

$$\begin{pmatrix} 10 & 3 \\ 1 & 9 \end{pmatrix} \otimes \begin{pmatrix} 4 & -5 \\ 27 & 0 \end{pmatrix} = \begin{pmatrix} 14 & 3 \\ 5 & -4 \end{pmatrix}$$

Let $t$ be an integer. If a matrix $A$ has the following form,

$$A = \begin{pmatrix} a_0 & a_{k-1} \otimes t & a_{k-2} \otimes t & \cdots & a_1 \otimes t \\ a_1 & a_0 & a_{k-1} \otimes t & \cdots & a_2 \otimes t \\ a_2 & a_1 & a_0 & \cdots & a_3 \otimes t \\ \vdots & \vdots & \vdots & \ddots & \vdots \\ a_{k-1} & a_{k-2} & a_{k-3} & \cdots & a_0 \end{pmatrix},$$

then it is called an upper $t$-circular matrix. We denote $A$ by $[a_0, a_1, \cdots, a_{k-1}]_k^t$ or $[a_0, a_1, \cdots, a_{k-1}]^t$. Let $C_k^t = \{A \in M_k(\mathcal{Z}) | A \text{ is upper } t-\text{circular matrix}\}$.

**Proposition 1.** *For any integer $t$, $C_k^t$ is a commutative sub-semiring of $M_k(\mathcal{Z})$.*

### 3. Public-Key Cryptography Using Tropical *T*-Circular Matrices
*3.1. Key Exchange Protocol Based on Tropical Circular Matrices*

**Definition 3.** *Let $s$ and $t$ be two integers. Let $P \in C_k^s$, $Q \in C_k^t$, and $Y \in M_k(\mathcal{Z}) \backslash (C_k^s \cup C_k^t)$. Suppose that $N = PYQ$. The two-side tropical circular matrix action problem (TCMAP) is to find two matrices $P \in C_k^s$, $Q \in C_k^t$ such that $N = PYQ$, given the matrices $N$ and $Y$.*

**Protocol 1.** *Let $k, s, t$ be three positive integers. Let $Y \in M_k(\mathcal{Z}) \backslash (C_k^s \cup C_k^t)$. In addition, $k, s, t$ and $Y$ are public.*

(1) *Alice selects at random two matrices $P_1 \in C_k^s$ and $Q_1 \in C_k^t$, and computes $K_a = P_1 Y Q_1$. In addition, she sends to Bob the matrix $K_a$.*
(2) *Bob selects at random two matrices $P_2 \in C_k^s$ and $Q_2 \in C_k^t$, and computes $K_b = P_2 Y Q_2$. He sends to Alice the vector $K_b$.*
(3) *Alice computes $K = P_1 K_b Q_1$. In addition, Bob computes $K = P_2 K_a Q_2$.*

*Since $C_k^s$ and $C_k^t$ are commutative sub-semirings of $M_k(\mathcal{Z})$, we have $P_1 P_2 = P_2 P_1$, $Q_1 Q_2 = Q_2 Q_1$ and*

$$P_1 K_b Q_1 = P_1 (P_2 Y Q_2) Q_1 = (P_1 P_2) Y (Q_2 Q_1) = (P_2 P_1) Y (Q_1 Q_2) = P_2 (P_1 Y Q_1) Q_2 = P_2 K_a Q_2$$

*Then, Alice and Bob share a secret key $K$.*

**Definition 4.** *Let $k, s, t$ be three positive integers. Let $P_1, P_2 \in C_k^s$, $Q_1, Q_2 \in C_k^t$ and $Y \in M_k(\mathcal{Z}) \backslash (C_k^s \cup C_k^t)$. Suppose that $K_a = P_1 Y Q_1$ and $K_b = P_2 Y Q_2$. The computational two-side tropical circular matrix action problem (CTCMAP) is to find a matrix $K \in M_k(\mathcal{Z})$ such that $K = P_1 P_2 Y Q_1 Q_2$, given the matrices $K_a, K_b$ and $Y$.*

**Proposition 2.** *An algorithm that solves TCMAP can be used to solve CTCMAP.*

**Theorem 1.** *Finding the common secret key from the public information of Protocol 1 is equivalent to solving CTCMAP.*

We give a practical example of Protocol 1 with small parameters in Appendix A.

**Remark 1.** *Protocol 1 is simplified. It can only resist passive attacks, but not active attacks, such as intruder-in-the-middle attacks. To avoid these attacks, it is desirable to have a procedure that authenticates Alice and Bob's identities to each other while the key is being formed. A standard way to stop an intruder-in-the-middle attack is the station-to-station (STS) protocol, which uses digital signatures.*

*The extended protocol makes use of certificates that, as usual, are signed by a TA (trusted authority). Each user U will have a signature scheme with a verification algorithm $\text{Ver}_U$ and a signing algorithm $\text{Sig}_U$. The TA also has a signature scheme with a public verification algorithm $\text{Ver}_{TA}$. Each user U has a certificate*

$$\textbf{Cert}(U) = (ID(U), \text{Ver}_U, \text{Sig}_{TA}(ID(U), \text{Ver}_U)),$$

*where ID(U) is certain identification information for U.*

**Protocol 2.** *The public domain parameters consist of $k, s, t$ and $Y$ as Protocol 1.*

*(1) Alice selects at random two matrices $P_1 \in C_k^s$ and $Q_1 \in C_k^t$, and computes $K_a = P_1 Y Q_1$. She sends **Cert**(A) and $K_a$ to Bob.*

*(2) Bob selects at random two matrices $P_2 \in C_k^s$ and $Q_2 \in C_k^t$, and computes*

$$K_b = P_2 Y Q_2, \ K = P_2 K_a Q_2 = P_2 P_1 Y Q_1 Q_2, \ y_b = \text{sig}_B(\text{ID(A)}||K_b||K_a).$$

*Then, Bob sends **Cert**(B), $K_b$ and $y_b$ to Alice.*

*(3) Alice verifies $y_b$ using $\text{Ver}_B$. If the signature $y_b$ is not valid, then she "rejects" and quits. Otherwise, she "accepts" and computes*

$$K = P_1 K_b Q_1 = P_1 P_2 Y Q_2 Q_1, \ y_a = \text{sig}_A(\text{ID(B)}||K_a||K_b),$$

*and she sends $y_a$ to Bob.*

*(4) Bob verifies $y_a$ using $\text{Ver}_A$. If the signature $y_a$ is not valid, then he "rejects"; otherwise, he "accepts".*

### 3.2. Public-Key Encryption Scheme Based on Tropical Circular Matrices

**Cryptosystem 1.**

(1) **Key generation:** Let $k, s, t$ be three positive integers. Let $P_1 \in C_k^s$, $Q_1 \in C_k^t$ and $Y \in M_k(\mathcal{Z}) \backslash (C_k^s \cup C_k^t)$. Suppose that $K_a = P_1 Y Q_1$. $k, s, t, Y$ are public. Alice's public key is $K_a$. Alice's secret key is $P_1, Q_1$.

(2) **Encryption:** Bob wants to send a message $M \in M_k(\mathbb{Z})$ to Alice.
  - (i)   Bob chooses at random $P_2 \in C_k^s$, $Q_2 \in C_k^t$ and computes $R = P_2 Y Q_2$ as a part of the ciphertext.
  - (ii)  Bob computes $S = M + P_2 K_a Q_2$ as the rest of the ciphertext, where "+" is the ordinary integer matrix addition.
  - (iii) Bob sends the ciphertext $(R, S)$ to Alice.

(3) **Decryption:** Alice receives the ciphertext $(R, S)$ and tries to decrypt it.
  - (i)  Using her secret key $P_1, Q_1$, Alice computes $T = P_1 R Q_1$.
  - (ii) Alice computes $S - T$, where "$-$" is the ordinary integer matrix subtraction.

Since

$$
\begin{aligned}
S - T \quad &= M + P_2 K_a Q_2 - P_1 R Q_1 \\
&= M + P_2 (P_1 Y Q_1) Q_2 - P_1 (P_2 Y Q_2) Q_1 \\
&= M + P_2 P_1 Y Q_1 Q_2 - P_1 P_2 Y Q_2 Q_1 \\
&= M + P_1 P_2 Y Q_1 Q_2 - P_1 P_2 Y Q_1 Q_2 \\
&= M,
\end{aligned}
$$

Alice obtains the plaintext messages $M$.

**Definition 5.** *Let $k, s, t$ be three positive integers. Let $P_1, P_2 \in C_k^s$, $Q_1, Q_2 \in C_k^t$ and $Y, E \in M_k(\mathcal{Z}) \backslash (C_k^s \cup C_k^t)$. Suppose that $K_a = P_1 Y Q_1$ and $K_b = P_2 Y Q_2$. The decisional two-side tropical circular matrix action problem (DTCMAP) is to decide whether $E = P_1 P_2 Y Q_1 Q_2$, given $Y, K_a, K_b, E$.*

**Proposition 3.** *An algorithm that solves CTCMAP can be used to solve DTCMAP.*

**Theorem 2.** *An algorithm that solves DTCMAP can be used to decide the validity of the ciphertexts of Cryptosystem 1, and an algorithm that decides the validity of the ciphertexts of Cryptosystem 1 can be used to solve DTCMAP.*

**Proof of Theorem 2.** Suppose first that the algorithm $\mathcal{A}_1$ can decide whether a decryption of Cryptosystem 1 is correct. In other words, when given the inputs $Y, K_a, (R, S), M$, the algorithm $\mathcal{A}_1$ outputs "yes" if $M$ is the decryption of $(R, S)$ and outputs "no" otherwise. Let us use $\mathcal{A}_1$ to solve the decisional two-side tropical circular matrix action problem. Suppose you are given $Y, K_a (= P_1 Y Q_1), K_b (= P_2 Y Q_2)$ and $E$, and you want to decide whether or not $E = P_1 P_2 Y Q_1 Q_2$. Let $K_a$ be the public key and $R = K_b$ be the first part of the ciphertext. Moreover, let $S = E$ be the second part of the ciphertext and $M = 0_{k \times k}$ be the zero matrix in $M_k(\mathcal{Z})$. Input all of these into $\mathcal{A}_1$. Note that, in the present setup, $P_1, Q_1$ are the secret keys. The correct decryption of $(R, S)$ is $S - P_1 R Q_1 = E - P_1 P_2 Y Q_1 Q_2$. Therefore, $\mathcal{A}_1$ outputs "yes" exactly when $M = 0$ is the same as $E - P_1 P_2 Y Q_1 Q_2$, namely, when $E = P_1 P_2 Y Q_1 Q_2$. This solves DTCMAP.

Conversely, suppose an algorithm $\mathcal{A}_2$ can solve DTCMAP. This means that if you give $\mathcal{A}_2$ inputs $Y, K_a (= P_1 Y Q_1), K_b (= P_2 Y Q_2)$ and $E$, then $\mathcal{A}_2$ outputs "yes" if $E = P_1 P_2 Y Q_1 Q_2$ and outputs "no" if not. Let $M$ be the claimed decryption of the ciphertext $(R, S)$. Input the public key $K_a$ and input $R = P_2 Y Q_2$ as $K_b$. Input $S - M$ as $E$.

Note that $M$ is the correct plaintext for the ciphertext $(R, S)$ if and only if $M = S - P_1 R Q_1 = S - P_1 P_2 Y Q_1 Q_2$, which happens if and only if $S - M = P_1 P_2 Y Q_1 Q_2$. Therefore, $M$ is the correct plaintext if and only if $E = P_1 P_2 Y Q_1 Q_2$. Therefore, with these inputs, $\mathcal{A}_2$ outputs "yes" exactly when $M$ is the correct plaintext. $\square$

## 4. Security and Parameter Selection

Through Theorem 1, Proposition 3, and Theorem 2, an efficient algorithm for solving the two-side tropical circular matrix action problem can be used to attack Protocol 1 and Cryptosystem 1.

**Proposition 4.** *TCMAP can be reduced to the problem of solving a tropical nonlinear system of equations.*

**Proof of Proposition 4.** Let $P \in C_k^s$, $Q \in C_k^t$ and $Y \in M_k(\mathcal{Z}) \backslash (C_k^s \cup C_k^t)$. Suppose that $N = PYQ$. Now, we can try to find two matrices, $P \in S_1$ and $Q \in S_2$, such that $N = PYQ$, given $N$ and $Y$.

Suppose that $P = [x_0, x_1, \cdots, x_{k-1}]^s$ and $Q = [y_0, y_1, \cdots, y_{k-1}]^t$. Then,

$$
[x_0, x_1, \cdots, x_{k-1}]^s \cdot Y \cdot [y_0, y_1, \cdots, y_{k-1}]^t = N
$$

Since $Y$ and $N$ are known, we obtain a tropical nonlinear system of equations about $x_0, x_1, \cdots, x_{k-1}, y_0, y_1, \cdots, y_{k-1}$ with $2k$ unknowns and $k^2$ equations. $\square$

As we know, the problem of solving a tropical nonlinear system of equations is usually NP-hard [24]. We present an algorithm for solving the two-side tropical circular matrix action problem with exponential computational complexity.

**Proposition 5.** *There exists an algorithm for solving the two-side tropical circular matrix action problem with computational complexity* $O\left( k^4 + 6k^3 \cdot \binom{k^2}{2k} \right)$.

**Proof of Proposition 5.** With Proposition 4, we obtain a tropical nonlinear system of equations about $x_0, x_1, \cdots, y_{k-1}$ with $2k$ unknowns and $k^2$ equations. Note that every term of the equations is the form of $x_i y_j$ $(i, j = 0, 1, \cdots, k-1)$. Denote $z_0 = x_0 y_0$, $z_1 = x_0 y_1, \cdots, z_{k^2} = x_{k-1} y_{k-1}$. Then, we obtain a tropical linear system of equations with $k^2$ unknowns $z_i$ and $k^2$ equations.

After solving the tropical linear system of equations of $z_i$, we can obtain a system of nonlinear equations

$$x_0 y_0 = z_0, x_0 y_1 = z_1, \cdots, x_{k-1} y_{k-1} = z_{k^2}$$

Since multiplication in tropical algebra is an ordinary addition, it is actually a system of linear equations over an integer ring. The linear equations have $2k$ unknowns and $k^2$ equations. Generally, the system of linear equations has no solution. However, if the $2k$ equations in these $k^2$ equations have a solution, it is possible to find $x_0, x_1, \cdots, y_{k-1}$ such that

$$[x_0, x_1, \cdots, x_{k-1}]^s \cdot Y \cdot [y_0, y_1, \cdots, y_{k-1}]^t = N.$$

Using the algorithm in [33], the complexity of solving the tropical linear system of equations with $k^2$ unknowns $z_i$ and $k^2$ equations is $O(k^4)$. The number of possible choices for selecting $2k$ equations from $k^2$ equations is $\binom{k^2}{2k}$. The complexity of solving integer linear equations with $2k$ equations and $2k$ unknowns is $O((2k)^3)$. Therefore, the computational complexity of the above algorithm is $O\left( k^4 + 6k^3 \cdot \binom{k^2}{2k} \right)$. $\square$

An example of solving TMCAP with small parameters is given in Appendix B.

*4.1. KU Attack*

Because the commutative semiring used in our cryptosystems is the semiring of all $t$-circular matrices, this is different from that of Grigoriev and Shpilrain's public-key cryptosystem I [24]. They used two public tropical matrices $M_1, M_2$ and $(M_1 M_2 \neq M_2 M_1)$ and then adopted the commutative semiring $\mathcal{Z}[M_1]$, $\mathcal{Z}[M_2]$. Let $p_1(M_1) \in \mathcal{Z}[M_1]$, $p_2(M_2) \in \mathcal{Z}[M_2]$ and $p_1(M_1) Y p_2(M_2) = U$. The security of their cryptosystem relies on the difficulty of the problem of finding $S_1 \in \mathcal{Z}[M_1]$ and $S_2 \in \mathcal{Z}[M_2]$ such that $S_1 Y S_2 = U$. (Note that $S_1$ may not be equal to $p_1(M_1)$ and $S_2$ may not be equal to $p_2(M_2)$.) Because the secret matrix can be represented by a polynomial of $M_1, M_2$, Kotov and Ushakov [25] designed an efficient algorithm to attack the key exchange protocol in [24]. Suppose that

$$S_1 = \sum_{i=0}^{D} x_i M_1^i, \ S_2 = \sum_{i=0}^{D} y_i M_2^i,$$

where unknowns $x_i, y_j \in \mathcal{Z}$, and $D$ is the upper bound for the degree of polynomials. $S_1 Y S_2 = U$ gives $\sum_{i=0}^{D} x_i y_j M_1^i Y M_2^j = U$. This translates to

$$\min(x_i + y_j + T_{rs}^{ij}) = 0, \ \forall 1 \leq r, s \leq k$$

where $T^{ij} = M_1^i Y M_2^j - U$. A specific description of KU attack is presented as Algorithm 1.

---
**Algorithm 1:** KU Attack algorithm

---
Input: $M_1, M_2, U(= p_1(M_1)Yp_2(M_2))$.

Output: $x_1, \cdots, x_D, y_1, \cdots, y_D$, such that $S_1 Y S_2 = U$, where $S_1 = \sum_{i=0}^{D} x_i M_1^i$, $S_2 = \sum_{i=0}^{D} y_i M_2^i$.

(1) Compute $m_{ij} = \min_{i,j}(T_{rs}^{ij})$ and $P_{ij} = \left\{ (r,s) \middle| T_{rs}^{ij} = m_{ij} \right\}$;

(2) Among all minimal covers of $\{1, 2, \cdots, k\} \times \{1, 2, \cdots, k\}$ by $P_{ij}$, that is, all minimal subsets $C \subseteq \{0, 1, \cdots, D\} \times \{0, 1, \cdots, D\}$ such that

$$\underset{(i,j) \in C}{\cup} P_{ij} = \{1, 2, \cdots, k\} \times \{1, 2, \cdots, k\}$$

find a cover for which the system

$$\begin{cases} x_i + y_j = -m_{ij}, & \text{if } (i,j) \in C \\ x_i + y_j \geq -m_{ij}, & \text{if } (i,j) \notin C \end{cases}$$

is solvable.

---

Experimental results show that the attack algorithm can succeed in a short amount of time when the parameters are small ($k \leq 40$, $D \leq 40$, and the entries of matrices and the coefficients of polynomials are integers in $[-10^{10}, 10^{10}]$).

Since tropical $t$-circular matrices cannot be represented by a known matrix, our cryptosystem can resist KU attacks.

*4.2. RM Attacks*

Grigoriev and Shpilrain [26] improved the original scheme and proposed a public-key cryptosystem based on the semidirect product of the tropical matrix semiring. Let $S = (M_k(\mathcal{Z}), \oplus, \otimes)$ be the tropical semiring of $k \times k$ tropical matrices over $\mathcal{Z}$. It can be seen that $S \times S$ is a semigroup under the operation $\circ$ given as

$$(\forall (M_1, H_1), (M_2, H_2) \in S \times S)$$
$$(M_1, H_1) \circ (M_2, H_2) = ((M_1 \oplus H_2 \oplus M_1 \otimes H_2) \oplus M_2, H_1 \oplus H_2 \oplus H_1 \otimes H_2).$$

Using the semigroup $(S \times S, \circ)$, Grigoriev and Shpilrain proposed an improved tropical public-key cryptosystem. However, cryptanalysis of the improved tropical public-key cryptosystem was successfully implemented using a simple binary search by Rudy and Monico [27]. A partial order on $S$ is defined as

$$(\forall X, Y \in S) X \leq Y \text{ if } x_{ij} \leq y_{ij} \ \forall i, j \in \{1, \cdots, k\}.$$

It can be easily observed that for the operations $\circ$, if $(M, H)^p$ is denoted by $(M_p, H_p)$, then the sequence $\{M_p\}$ is monotonically decreasing, i.e., $M_1 \geq M_2 \geq M_3 \geq \cdots$ and so on. Algorithm 2 gives the pseudocode description of RM attack.

---
**Algorithm 2:** RM Attack algorithm

---
Input: $M, H, A \in S$, where $(M, H)^m = (A, H^m)$, for some positive integer $m$ ($1 \leq m \leq r$).
Output: $m$.
(1) Let $left = 1$ and $right = r$;
(2) Execute the following loop when $left \leq right$.
  (i) $mid = left + (right - left)/2$
  (ii) Compute $(M, H)^{mid} = (P, Q)$.
    If $P < A$, $right = mid - 1$;
    If $P > A$, $left = mid + 1$;
    If $P = A$, output $m = mid$.

---

In our cryptosystems, there is no tropical matrix addition operation $\oplus$ and the partial order cannot be used. Thus, our cryptosystems can resist RM attacks. We compare the

security among relevant cryptosystems in [24,26] and our proposed cryptosystem. The comparison results are depicted in Table 1.

**Table 1.** Comparison among relevant tropical schemes.

| Schemes | Mathematical Problems | KU Attack | RM Attack |
|---|---|---|---|
| Grigoriev et al. [24] | Two-side matrix action problem | × | √ |
| Grigoriev et al. [26] | Semidirect product problem | √ | × |
| Our scheme | Two-side tropical circular matrix action problem | √ | √ |

Note that √ means that the scheme can resist the corresponding attack, while × means it does not.

### 4.3. Parameter Selection

Table 2 shows the performance comparison of the cryptosystem under some different parameters, where the entries of the matrices are integers in $[0, 2^{64})$.

**Table 2.** Performance comparison under some different parameters.

| $k$ | Size of sk (kB) | Size of pk (kB) | Complexity of Solving TCMAP |
|---|---|---|---|
| 10 | 0.0781 | 0.7813 | $O(2^{81})$ |
| 20 | 0.1563 | 3.1250 | $O(2^{199})$ |
| 30 | 0.2344 | 7.0313 | $O(2^{331})$ |
| 40 | 0.3125 | 12.5000 | $O(2^{472})$ |
| 50 | 0.3906 | 19.5313 | $O(2^{620})$ |
| 60 | 0.4688 | 28.1250 | $O(2^{775})$ |

Note that "sk" means secret key and "pk" means public key.

In Table 3, we list the computation time for related cryptographic operations in our cryptosystem on different platforms, where $k = 50$, $s = t = 100101$, and the entries of the matrices are integers in $[0, 2^{64})$.

**Table 3.** Timings for cryptographic operations in our cryptosystem.

| Experimental Platform | Key Generation | Encryption | Decryption |
|---|---|---|---|
| Intel (R) i7-8550 1.80 GHz | 0.984 s | 1.018 s | 0.513 s |
| Intel (R) i5-5200 2.20GHz | 0.624 s | 0.594 s | 0.297 s |
| Intel (R) i7-4700 2.40GHz | 0.363 s | 0.346 s | 0.187 s |

We recommend using the parameters $k \geq 50$, $s, t \in (0, 2^{32})$, and the entries of the matrices of integers in $[0, 2^{64})$ to avoid potential heuristic attacks similar to KU attacks.

## 5. Conclusions and Further Research

In this paper, we present a new key exchange protocol and a new public-key encryption scheme based on tropical matrices. We use a class of tropical commuting matrix, that is, the tropical $t$-circular matrix, other than matrix powers or matrix polynomials. The security of new public-key cryptosystems relies on a two-side tropical circular matrix action problem (TCMAP). The use of $t$-circular matrices allows us to share less information with the attacker. Since tropical circular matrices cannot be represented by a known matrix, our public-key cryptosystems can resist KU attacks. There is no addition of tropical matrices in our schemes. So, the attack method proposed by Rudy and Monico does not work for our public-key cryptosystems. Our public-key cryptosystem can resist all known attacks. As we know, the best way to solve TCMAP is to solve a tropical nonlinear system of equations, which is NP-hard. So, the new cryptosystems can be considered as a potential post-quantum cryptosystem.

Future works worth studying include the following:

(1) A possible algorithm for solving TCMAP. If we can find some algorithms for solving the systems of min-plus polynomial equations, then they can be used to attack our schemes.

(2) Other cryptographic applications of TCMAP. For example, we can try to design digital signature schemes and identity authentication schemes based on TCMAP.

**Author Contributions:** Conceptualization, H.H. and C.L.; methodology, H.H. and L.D.; software, H.H. and L.D.; validation, H.H. and L.D.; formal analysis, H.H., C.L. and L.D.; writing—original draft preparation, H.H. and L.D.; writing—review and editing, H.H., C.L. and L.D. All authors have read and agreed to the published version of the manuscript.

**Funding:** This work is supported by the National Natural Science Foundation of China (No. 61962011, 61462016) and the Science and Technology Foundation of Guizhou Province (QIANKEHEJICHU-ZK [2021] 313).

**Institutional Review Board Statement:** Not applicable.

**Informed Consent Statement:** Not applicable.

**Data Availability Statement:** Not applicable.

**Conflicts of Interest:** The authors declare no conflict of interest.

### Notations

In this paper, the matrix is generally denoted by capital letters. Frequently used notations are listed below with their meanings:

| | |
|---|---|
| $\mathbb{Z}$ | set of integers; |
| $\mathcal{Z}$ | tropical semiring of integers $\mathbb{Z} \cup \{\infty\}$; |
| $M_k(\mathcal{Z})$ | set of all $k \times k$ tropical matrices over $\mathcal{Z}$; |
| $C_k^t$ | set of all $k \times k$ tropical upper $t$-circular matrices over $\mathcal{Z}$; |
| TCMAP | two-side tropical circular matrix action problem; |
| CTCMAP | computational two-side tropical circular matrix action problem; |
| DTCMAP | decisional two-side tropical circular matrix action problem. |

### Appendix A. An Example of Protocol 1 with Small Parameters

We choose the parameters $k = 5$ and $s = t = 9361$ and the entries of the matrices in $[0, 2^{15})$. The public matrix $Y$ is as follows:

$$Y = \begin{pmatrix} 8630 & 29,391 & 21,921 & 18,968 & 25,014 \\ 15,306 & 5461 & 18,973 & 800 & 1786 \\ 7986 & 27,430 & 22,510 & 11,233 & 30,900 \\ 2398 & 6071 & 25,269 & 27,186 & 4328 \\ 18,306 & 10,527 & 16,873 & 11,565 & 9569 \end{pmatrix},$$

(1) Alice selects at random two $t$-circular matrices $P_1, Q_1$ as follows:

$$P_1 = [297, 21, 730, 15, 290, 10, 135, 19, 522]^{9361}$$

$$Q_1 = [21, 654, 19, 077, 27, 810, 23, 876, 1267]^{9361}$$

Alice computes $K_a = P_1 Y Q_1$. She sends the matrix $K_a$ to Bob.

$$K_a = \begin{pmatrix} 26,578 & 19,555 & 38,342 & 32,846 & 29,893 \\ 3350 & 25,959 & 16,386 & 21,160 & 11,725 \\ 24,783 & 18,911 & 30,607 & 33,184 & 22,158 \\ 5892 & 13,323 & 16,996 & 23,702 & 26,279 \\ 11,133 & 29,231 & 21,452 & 27,798 & 21,563 \end{pmatrix}.$$

(2)　Bob selects at random two $t$-circular matrices $P_2, Q_2$ as follows:

$$P_2 = [1059\ 4901\ 20,575\ 21,400\ 4378]^{9361}$$

$$Q_2 = [8556\ 14,895\ 30,549\ 31,378\ 15,257]^{9361}$$

Bob computes $K_b = P_2 Y Q_2$. He sends the matrix $K_b$ to Alice.

$$K_b = \begin{pmatrix} 18,245 & 27,756 & 29,434 & 23,095 & 24,081 \\ 18,102 & 15,076 & 16,754 & 10,415 & 11,401 \\ 17,601 & 18,918 & 20,596 & 14,257 & 15,243 \\ 12,013 & 15,686 & 31,029 & 20,282 & 13,943 \\ 15,855 & 19,528 & 26,488 & 21,180 & 17,785 \end{pmatrix}.$$

(3)　Alice computes $K = P_1 K_b Q_1$. Bob computes $K = P_2 K_a Q_2$.

$$K = \begin{pmatrix} 25,645 & 29,170 & 38,681 & 40,359 & 34,020 \\ 12,965 & 29,027 & 26,001 & 27,679 & 21,340 \\ 16,807 & 28,526 & 29,843 & 31,521 & 25,182 \\ 15,507 & 22,938 & 26,611 & 33,317 & 31,207 \\ 19,349 & 26,780 & 30,453 & 37,159 & 31,178 \end{pmatrix}.$$

**Appendix B. An Example of Solving TMCAP with Small Parameters**

We choose the parameters $k = 3$ and $s = t = 23$ and the entries of the matrices in $[0, 100]$. The public matrix $Y$ is as follows:

$$Y = \begin{pmatrix} 81 & 24 & 82 \\ 5 & 52 & 98 \\ 3 & 2 & 69 \end{pmatrix},$$

Alice selects at random two $t$-circular matrices $P_1, Q_1$ as follows:

$$P_1 = [0\ 8\ 31]^{23}, \ Q_1 = [68\ 0\ 6]^{23}.$$

Alice computes $K_a = P_1 Y Q_1$. She sends the matrix $K_a$ to Bob.

$$K_a = \begin{pmatrix} 24 & 63 & 53 \\ 32 & 34 & 28 \\ 2 & 32 & 26 \end{pmatrix},$$

The attacker knows $k, t, Y$ and obtains $K_a$. They try to find $P_1$ and $Q_1$.
Let $P_1 = [x_0\ x_1\ x_2]^{23}$ and $Q_1 = [y_0\ y_1\ y_2]^{23}$. Then,

$$[x_0\ x_1\ x_2]^{23} \begin{pmatrix} 81 & 24 & 82 \\ 5 & 52 & 98 \\ 3 & 2 & 69 \end{pmatrix} [y_0\ y_1\ y_2]^{23} = \begin{pmatrix} 24 & 63 & 53 \\ 32 & 34 & 28 \\ 2 & 32 & 26 \end{pmatrix} (\sharp).$$

From it, they can obtain the tropical linear equations,

$$
\left\{
\begin{array}{rcl}
81x_0y_0 \oplus 24x_0y_1 \oplus 82x_0y_2 \oplus 26x_1y_0 \oplus 25x_1y_1 \oplus 92x_1y_2 \oplus 28x_2y_0 \oplus 75x_2y_1 \oplus 121x_2y_2 & = & 24 \\
24x_0y_0 \oplus 82x_0y_1 \oplus 124x_0y_2 \oplus 25x_1y_0 \oplus 92x_1y_1 \oplus 49x_1y_2 \oplus 75x_2y_0 \oplus 121x_2y_1 \oplus 51x_2y_2 & = & 63 \\
82x_0y_0 \oplus 124x_0y_1 \oplus 47x_0y_2 \oplus 92x_1y_0 \oplus 49x_1y_1 \oplus 48x_1y_2 \oplus 121x_2y_0 \oplus 51x_2y_1 \oplus 98x_2y_2 & = & 53 \\
5x_0y_0 \oplus 52x_0y_1 \oplus 98x_0y_2 \oplus 81x_1y_0 \oplus 24x_1y_1 \oplus 82x_1y_2 \oplus 26x_2y_0 \oplus 25x_2y_1 \oplus 92x_2y_2 & = & 32 \\
52x_0y_0 \oplus 98x_0y_1 \oplus 28x_0y_2 \oplus 24x_1y_0 \oplus 82x_1y_1 \oplus 124x_1y_2 \oplus 25x_2y_0 \oplus 92x_2y_1 \oplus 49x_2y_2 & = & 34 \\
98x_0y_0 \oplus 28x_0y_1 \oplus 75x_0y_2 \oplus 82x_1y_0 \oplus 124x_1y_1 \oplus 47x_1y_2 \oplus 92x_2y_0 \oplus 49x_2y_1 \oplus 48x_2y_2 & = & 28 \\
3x_0y_0 \oplus 2x_0y_1 \oplus 69x_0y_2 \oplus 5x_1y_0 \oplus 52x_1y_1 \oplus 98x_1y_2 \oplus 82x_2y_0 \oplus 24x_2y_1 \oplus 82x_2y_2 & = & 2 \\
2x_0y_0 \oplus 69x_0y_1 \oplus 26x_0y_2 \oplus 52x_1y_0 \oplus 98x_1y_1 \oplus 28x_1y_2 \oplus 24x_2y_0 \oplus 82x_2y_1 \oplus 125x_2y_2 & = & 32 \\
69x_0y_0 \oplus 26x_0y_1 \oplus 25x_0y_2 \oplus 98x_1y_0 \oplus 28x_1y_1 \oplus 75x_1y_2 \oplus 82x_2y_0 \oplus 125x_2y_1 \oplus 47x_2y_2 & = & 26
\end{array}
\right.
$$

where $ax_iy_j$ denotes $a \otimes x_i \otimes y_j$. After solving the tropical linear equations, the attacker can obtain a solution, for example:

$$
\left\{
\begin{array}{rcll}
x_0 \otimes y_0 & = & 39 & \text{(A1)} \\
x_0 \otimes y_1 & = & 0 & \text{(A2)} \\
x_0 \otimes y_2 & = & 6 & \text{(A3)} \\
x_1 \otimes y_0 & = & 38 & \text{(A4)} \\
x_1 \otimes y_1 & = & 8 & \text{(A5)} \\
x_1 \otimes y_2 & = & 14 & \text{(A6)} \\
x_2 \otimes y_0 & = & 9 & \text{(A7)} \\
x_2 \otimes y_1 & = & 7 & \text{(A8)} \\
x_2 \otimes y_2 & = & 12 & \text{(A9)}
\end{array}
\right.
$$

where "+" denotes the ordinary addition.

It is easy to verify that (A1)–(A6) have no solution. (A2)–(A7) also have no solution.

The attacker keeps looking for a combination that may have a solution until they find a combination that has a solution. For example, they find that combinations (A1)–(A3), (A5), (A6), and (A8) have a solution $x_0 = 0, x_1 = 8, x_2 = 7, y_0 = 39, y_1 = 0, y_2 = 6$. The attacker substitutes this solution into ($\sharp$) to verify that it is a true solution of ($\sharp$). An attacker can find a solution by trying, at most, $\binom{9}{6}$ cases.

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
