# Peer review of "Public-Key Cryptography Based on Tropical Circular Matrices"

_applsci, doi:10.3390/app12157401_

Round 1
Reviewer 1 Report
I read this technical note; with a lot of interest. Mainly, it discusses the use public-key cryptography based on tropical circular matrices. Honestly speaking, though I am intrigued by all the definition mentioned in this paper and found that this work is very superfluous. In fact most the definition is well know which provides nothing new to the reader. I would expect that the authors provide some conclusions at the end of the paper, so that the authors understand the message of the paper.
Nevertheless, the following section could be improved:
1. The abstract needs to rewritten with more technical words. It is not addressing with given research topic.
2. Listing of all the abbreviation is important.
3. Introduction is very brief- A systematic and detail introduction is required.
4. Literature of current research is missing in my opinion.
5. Specify the research question and purpose of this study.
6. Main contribution of this research to the existing literature is missing? Inclusion is highly reccomemded.
7. Discussion section is very poorly written. Improvement is strongly suggested.
8. Is there software simulation to verify the outcomes?
9. No conclusion is provided. Inclusion of a scientific conclusion of this work will be beneficial for the readers.
10. Very limited references are provided. Recommendation to provide more extensive references.
Overall, the paper in its current form is more of concept paper that is appropriate for a conference rather than a journal paper in the peer reviewed Applied Science. I am sorry to say that though the paper has a good idea. I would suggest authors to give some more depth in their paper, some points that inspire researchers and then resubmit to this journal.
Reviewer 2 Report
The authors proposed and designed a new public key cryptosystem based on tropical circular matrices and a key exchange protocol. The authors evaluated the security validation through the NP-hardness of the problem of solving a tropical nonlinear system of equations. The authors considered the model robustness through the attacks such as the Kotov-Ushakov attack. This proposed model is interesting, and it is useful to enhance the security of systems, however, the paper has some shortcomings regarding some text that the authors are required to respond to. Here are my comments and suggestions.
- The context of the manuscript has shortcomings, the authors must modify that.
- In section (2), there are some parts that seem they remain from the template, it is not clear why the authors kept them, the authors need to explain about, line 59 to 65 and 80 to 82 and 86 to 88?
- Line 78, are zero element and unitary element defined correctly?
- Line 105, Protocol 3.2. explained about the shared secret key between Alice and Bob. If Eve (adversary) uses her selected two matrices and computes Keve and sends to Bob the matrix Keve, then how Bob finds out this attack?
- Line 116, Definition 3.3: is not well-defined.
- Line 179, Security and Parameter Selection, needs to extend to explain the attacks which were considered in this model. Furthermore, this section needs to improve.
Lots of abbreviations were used without giving the full word:
- Line 229: the RM is abbreviation of which attack?
- Line 256: CTCMA?
- Line 257: DTCMA?
- Line 281: Bob sends matrix Ka to Alice, shouldn’t be matrix Kb which Bob sends to ALice?
Round 2
Reviewer 1 Report
Revision is convincing!
Reviewer 2 Report
The authors have modified the manuscript and removed shortcomings.